# Dating *Alphaproteobacteria* evolution with eukaryotic fossils

Sishuo Wang [1] & Haiwei Luo [1,2,3✉]

Elucidating the timescale of the evolution of *Alphaproteobacteria*, one of the most prevalent microbial lineages in marine and terrestrial ecosystems, is key to testing hypotheses on their co-evolution with eukaryotic hosts and Earth's systems, which, however, is largely limited by the scarcity of bacterial fossils. Here, we incorporate eukaryotic fossils to date the divergence times of *Alphaproteobacteria*, based on the mitochondrial endosymbiosis that mitochondria evolved from an alphaproteobacterial lineage. We estimate that *Alphaproteobacteria* arose ~1900 million years (Ma) ago, followed by rapid divergence of their major clades. We show that the origin of *Rickettsiales*, an order of obligate intracellular bacteria whose hosts are mostly animals, predates the emergence of animals for ~700 Ma but coincides with that of eukaryotes. This, together with reconstruction of ancestral hosts, strongly suggests that early *Rickettsiales* lineages had established previously underappreciated interactions with uni-cellular eukaryotes. Moreover, the mitochondria-based approach displays higher robustness to uncertainties in calibrations compared with the traditional strategy using cyanobacterial fossils. Further, our analyses imply the potential of dating the (bacterial) tree of life based on endosymbiosis events, and suggest that previous applications using divergence times of the modern hosts of symbiotic bacteria to date bacterial evolution might need to be revisited.

[1] Simon F. S. Li Marine Science Laboratory, School of Life Sciences and State Key Laboratory of Agrobiotechnology, The Chinese University of Hong Kong, Shatin, SAR, Hong Kong. [2] Shenzhen Research Institute, The Chinese University of Hong Kong, Shenzhen, China. [3] Hong Kong Branch of Southern Marine Science and Engineering Guangdong Laboratory (Guangzhou), Guangzhou, SAR, Hong Kong. ✉email: hluo2006@gmail.com

The *Alphaproteobacteria* is one of the largest groups within bacteria[1] and of great evolutionary significance for holding the origin of the mitochondrion[2,3]. The *Alphaproteobacteria* has extensively diversified since its ancient origin, and comprise some of the most environmentally abundant and metabolically diverse organisms on Earth[1,4,5]. *Alphaproteobacteria* represent 40–50% of bacterioplankton cells in sunlit oceans and 20–30% in dark oceans[6,7], and account for ~30% of the dominant phylotypes of bacteria in global soils[8]. Besides, the intimate association between some alphaproteobacterial lineages and eukaryotes is of central importance for agricultural (e.g., rhizobia) and medical (e.g., rickettsia) applications. This makes *Alphaproteobacteria* a promising system to study the timing of bacterial evolution and their correlation with geological, ecological, and evolutionary events[5,9,10]. However, the traditional way of divergence time estimation has several limits when applied to *Alphaproteobacteria* and other prokaryotes. First, fossil records of bacteria are extremely scarce and controversial in terms of their estimated dates[11]. Second, the most widely used prokaryotic fossils are from cyanobacteria, but the long evolutionary distance between cyanobacteria and other bacteria causes large uncertainties in dating[12]. Last, some studies[13,14] assumed a strict relationship of bacteria-host evolution and calibrated the evolution of symbiotic bacteria based on the divergence time of their modern hosts (mostly animals and plants). However, this precludes the possibility of host switching, which could occur frequently during evolutionary processes spanning millions of years[9]. Owing to these challenges, the origin time of *Alphaproteobacteria* estimated by previous studies varies from <600 million years (Ma) to >2000 Ma[5,14–16], making any narratives based on its evolutionary timing contentious.

Recently, horizontal gene transfer (HGT) has been suggested to have great potential in dating the evolution of bacteria[17,18]. In brief, if in an HGT event the recipient has fossil records while the donor does not, the temporal information recorded in the recipient can be transferred to date the evolution of the donor group (and vice versa), thereby bypassing the paucity of fossils in the donor lineage. Inspired by this idea, we develop a new strategy to date the divergence times of *Alphaproteobacteria* based on the mitochondrial endosymbiosis that the mitochondrion was derived from a bacterial lineage[19], whose phylogenetic position was later shown to be within[2,3,20] or closely related to[21] *Alphaproteobacteria* by modern phylogenetic analysis. As mitochondria are characteristic of eukaryotes, here we take advantage of eukaryotic fossils to anchor the divergence time of *Alphaproteobacteria* in a tree integrating both alphaproteobacterial and mitochondrial lineages.

## Results

### The evolutionary timescale of *Alphaproteobacteria* estimated by a mitochondria-based strategy.
We first reconstructed a phylogenomic tree of 80 carefully selected *Alphaproteobacteria* and mitochondrial genomes using 24 conserved genes based on prior phylogenomics studies[21,22] (see Methods and Supplementary Note 1.1). We employed rigorous approaches to delineate phylogenetic artefacts caused by long-branch attraction and compositional heterogeneity (see Methods), and obtained results consistent with recent studies where (i) *Rickettsiales*, *Holosporales*, and *Pelagibacterales* (SAR11) had independent origins[22], and (ii) mitochondria branched as a sister to *Alphaproteobacteria*[21] (Supplementary Fig. 1A). We also tested the impact of alternative topologies on dating (Supplementary Fig. 1C; see below). We compiled two data sets to estimate the time divergences within the *Alphaproteobacteria* calibrated by eukaryotic fossils with relaxed molecular clocks[23], which accounts for substitution rate

variations among branches. The first data set, which we referred to as the mito-encoded data set, was based on the aforementioned 24 conserved genes encoded by mitochondrial genomes (Supplementary Data 1), and the mitochondrial lineages mainly comprising species of green plants, red algae, and jakobids, whose mitochondrial genomes are both gene-rich and relatively slowly evolving[21,24]. Four high-confidence fossils from land plants and red algae were used as the calibration points (Supplementary Note 2.1; Supplementary Fig. 2A). The second data set, referred to as the nuclear-encoded data set, was based on 22 mitochondria-derived genes that had been transferred to the nuclear genome identified by Wang and Wu[3] (Supplementary Data 1). This data set not only circumvented the problem that in many eukaryotes genes encoded by mitochondrial genomes are few (e.g., apicomplexans and dinoflagellates) or fast-evolving (e.g., animals and fungi)[25], but integrated six additional eukaryotic fossils (Supplementary Fig. 2A; Supplementary Note 2.1), allowing directly comparing the divergence time between animal symbionts and their hosts on the same tree.

We selected a best-practiced scheme based on systematic comparisons of different combinations of parameters of MCMCTree for the mito- and nuclear-encoded data sets (see Methods and Supplementary Note 1.2). Similar divergence times were recovered for most nodes between the two data sets, although the mito-encoded data set estimated older ages for deep nodes (Fig. 1). As shown in the infinite-sites plots (up- and bottom-left panels in Fig. 1), the posterior mean ages versus 95% HPD (highest posterior density) widths approached a straight line, suggesting that the uncertainty in time estimate was predominantly caused by the uncertainty associated with fossil calibrations[26]. The estimated ages in the nuclear-encoded data set exhibited smaller 95% HPD intervals, hence smaller uncertainties, likely owing to their more calibration information compared with the mito-encoded data set.

Most alphaproteobacterial orders diverged 1500–1000 Mya, and *Rickettsiales* and *Pelagibacterales* appeared to be the oldest and youngest alphaproteobacterial orders, respectively, based on the taxa sampled here (Fig. 1). The origin time of *Rickettsiales*, an obligate endosymbiont lineage whose hosts cover diverse eukaryotes but mostly animals[27], was estimated to be 1741 Ma (95% HPD 1975–1514 Ma) and 1607 Ma (95% HPD 1767–1467 Ma) using the mito- and nuclear-encoded data sets, respectively (Fig. 1). One merit of our mitochondria-based method is that divergence times of the eukaryotic hosts and of the host-associated bacteria can be simultaneously estimated. As shown in Fig. 1, we dated the origin of animals (Metazoa) to be 814 Ma (95% HPD 850–769 Ma), consistent with previous dating analyses[28–30], but not others[31]. We also estimated that mitochondrial lineages diverged from *Alphaproteobacteria* ~1900 Mya (million years ago) and that the last common ancestor (LCA) of mitochondria occurred ~1550 Mya (Fig. 1). Thus, the origin of *Rickettsiales* likely predated the evolutionary emergence of animals for ~700 Ma but coincided with the mitochondrial endosymbiosis process and the occurrence of the LCA of eukaryotes, according to our (Fig. 1) and others' estimates[28,29,32], and fossil records (reviewed in Butterfield[33]). This agrees with recent findings of an increasingly broad range of protistan hosts of *Rickettsiales* (reviewed in Castelli et al[.34]), and suggests that host switches to animals from protists occurred later in the evolution of *Rickettsiales*. The origin time of *Holosporales*, another important endosymbiotic lineage in *Alphaproteobacteria* whose extant members are mostly endonuclear parasites of the ciliate *Paramecium*[35], dated back to 1360 Ma (95% HPD 1557–1168 Ma) or 1281 Ma (95% HPD 1429–1138 Ma), respectively, based on the mito- or nuclear-encoded data sets. This implies that the origin of *Holosporales* roughly coincided with

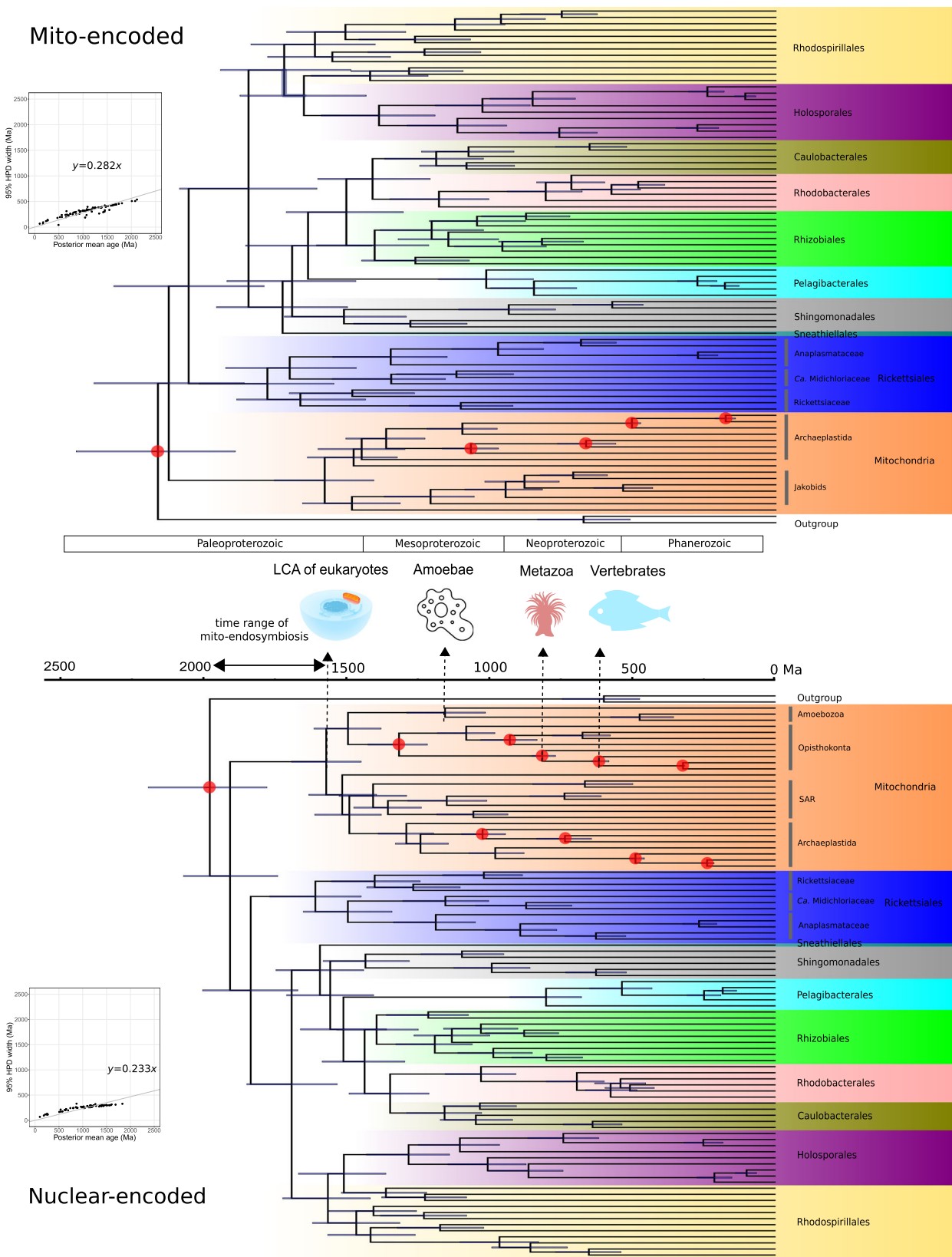

that of ciliates, which dated back to ~1150 Ma according to others' estimates[28,36]. Although the above analyses were based on amino-acid sequences using MCMCTree, the basic patterns held similar with PhyloBayes or using coding sequences, though PhyloBayes analysis based on the mito-encoded data set estimated older ages (Supplementary Fig. 3).

**Accommodating the uncertainties in the time estimation of *Alphaproteobacteria* evolution**. We assessed the impact of uncertainties in Bayesian relaxed molecular clock time estimation, including the disparity between fossil evidence and molecular clock estimates, root age, across-branch rate variation, sequence partitioning, and clock model, on the posterior ages

**Fig. 1 Divergence time estimate using the mitochondria-based strategy.** The evolutionary timeline of the *Alphaproteobacteria* estimated using the best-practiced dating scheme for the mito-encoded (the top) and nuclear-encoded (the bottom) data sets with *Magnetococcales* as the outgroup. Node bars denote the 95% HPD interval of estimated dates. Nodes with red circles denote calibration points. The organism images above the timeline indicate the origin times of eukaryotic lineages that *Rickettsiales* and *Holosporales* are mostly associated with, inferred simultaneously with *Alphaproteobacteria* using the same data set. The potential time range of mitochondrial endosymbiosis is also indicated. The panels in the upper and lower left corners are the infinite-sites plots, where the uncertainty in the divergence time (measured as the 95% HPD width) is plotted against the posterior mean of estimated times for each node. A lower value of the slope indicates less changes in the 95% HPD width, hence higher precision in dating. The image of mitochondria is credited to artist Kevin Song under CC BY-SA 3.0 (https://creativecommons.org/licenses/by-sa/3.0/) with slight modifications. All other organism images are distributed under CC0. The mito- and nuclear-encoded data sets have 5250 and 4936 sites, respectively. Calibrations for the mito-encoded data set: crown group angiosperms 250–125 Ma, crown group land plants 509–450 Ma, total group Florideophyceae 1891–550 Ma, total group red algae 1891–1033 Ma, root 3000–1000 Ma. Calibrations for the nuclear-encoded data set: crown group angiosperms 250–125 Ma, crown group land plants 509–450 Ma, total group Florideophyceae 1891–550 Ma, total group red algae 1891–1033 Ma, crown group Foraminifera 1891–525 Ma, crown group Amniota 332–318 Ma, crown group Chordata 636–520 Ma, crown group Metazoa 833–550 Ma, total group fungi 1891–890 Ma, crown group Dikarya 1891–400 Ma, root 3000–1000 Ma. A uniform distribution ranging from the minimum to maximum bound is applied for each calibration point. Both maximum and minimum bounds are soft, meaning that there is a small probability (2.5% by default) that the age is beyond the bound (see also Supplementary Note 2.1).

(Supplementary Data 2). When we used only Phanerozoic fossils and excluded all Proterozoic fossils (which are thought controversial by some), the estimated ages of most nodes were shifted towards the present by ~20% for the mito-encoded data set and ~12% for the nuclear-encoded data set (*Phan* in Fig. 2A, Supplementary Fig. 4). Removing potentially controversial maximum age constraints led to minor changes (<10%) in the posterior ages (*Max-1* and *Max-2* in Fig. 2A, Supplementary Fig. 4). Using more conservative calibrations of the root showed very similar time estimates (*Root-1*, *Root-2*, and *Root-3* in Supplementary Fig. 5). Decreasing the number of partitions resulted in decreased precision, as indicated by the increase of the slope in infinite-sites plots (Supplementary Fig. 6), but the estimated dates remained similar (Single partition in Fig. 2A). The largest changes in the posterior ages were obtained when the independent rates (IR) instead of autocorrelated rates clock model was used: the divergence times of most alphaproteobacterial orders were shifted towards the present by ~20% (IR in Fig. 2A, Supplementary Fig. 4). Collectively, the composite of the ages estimated from six different analyses shows that *Alphaproteobacteria* originated 1926 Ma (95% HPD 2423–1419 Ma) and 1748 Ma (95% HPD 2064–1424 Ma), based on the mito- and nuclear-encoded data sets, respectively (Supplementary Fig. 7), and diversified soon thereafter.

The phylogeny of *Alphaproteobacteria* is another much-debated issue[3,20,22,37]. We repeated the MCMCTree analysis by fixing the species phylogeny to 11 alternative topologies (Supplementary Fig. 1C). Most alphaproteobacterial orders showed highly consistent estimated ages (Fig. 2B). However, the posterior mean ages of *Holosporales* varied from ~2000 to ~1300 Ma across different topologies. This was because the alternative phylogenetic position of *Holosporales* was a sister to *Rickettsiales*, in contrast to the topology used in the main analysis where *Holosporales* branched within the *Rhodospirillales* (Supplementary Fig. 1C). Likewise, the origin time of *Pelagibacterales* showed considerable variations depending on whether it formed a monophyletic group with the *Rickettsiales*. Further topology tests with five statistical approaches, in general, rejected these alternative tree topologies with the only exception of topology 4 (Supplementary Table 3; Supplementary Fig. 1C), which represents a classical view of the phylogenetic position of mitochondria (the sister to *Rickettsiales*)[38]. Divergence times obtained based on topology 4 showed congruent results with the one used in the main analysis (topology 1 in Fig. 2B). Besides, we performed MCMCTree analysis with a wider taxonomic sampling by including additional 16 metagenome-assembled genomes, many of which are early-split alphaproteobacterial lineages not represented by the genomes used in the main analysis

(Supplementary Note 1.2). In general, this analysis returned similar time estimates compared with those obtained by the best-practiced dating scheme (Supplementary Fig. 8). Additional analyses that co-estimated both time and tree topology with BEAST showed different tree topologies but similar estimated ages compared with those estimated by MCMCTree where the tree topology was fixed (Supplementary Fig. 9). The difference in the tree topology is not surprising, as the methods and models employed by the two software are different. For example, the amino-acid sequence recoding and mixture model, which was used to generate the best-practiced tree in our phylogenomic reconstruction, are not implemented in BEAST. On the other side, the impact of the topology of the eukaryotic tree (mitochondria subtree) was relatively minor, as different topologies showed highly similar time estimates of alphaproteobacterial lineages at the order level (Supplementary Fig. 10).

Mitochondrial genes are known to be fast-evolving[21,24], causing obvious differences in the substitution rate between mitochondria and bacterial lineages (except for parasitic lineages like *Rickettsiales*[39]). Although in theory the violation of the molecular clock can be accommodated by relaxed molecular clock algorithms[23], it is necessary to assess its impact on the dating results. We classified genes into different categories according to the differences in substitution rate between mitochondria and non-*Rickettsiales Alphaproteobacteria*, and re-ran MCMCTree analysis based on genes of different rate categories. As shown in Supplementary Fig. 11, there appeared no apparent bias in estimated ages toward genes evolving at a more different rate between mitochondria and *Alphaproteobacteria*, suggesting that among-branch rate differences were well accounted for by relaxed molecular clock analyses and did not have a large impact on our analysis. In addition, allowing larger among-branch rate variation obtained highly consistent results (*Sigma* in Supplementary Fig. 5).

**The mitochondria-based strategy reduces dating uncertainty compared with the cyanobacteria-based method.** Traditionally, constructing the evolutionary timeline for bacteria is based on fossils of cyanobacteria since bacterial fossils that can be accurately assigned to a taxonomic group are only available for cyanobacteria[11]. Molecular dating commonly requires at least one calibration that provides a maximum age[40]. Fossils, by themselves, only provide the minimum bound. When there is no suitable maximum age for internal nodes, such as the case of cyanobacteria, the maximum constraint is typically provided at the root[40], which could vary a lot based on different evidence and in different studies, however. To accommodate this uncertainty,

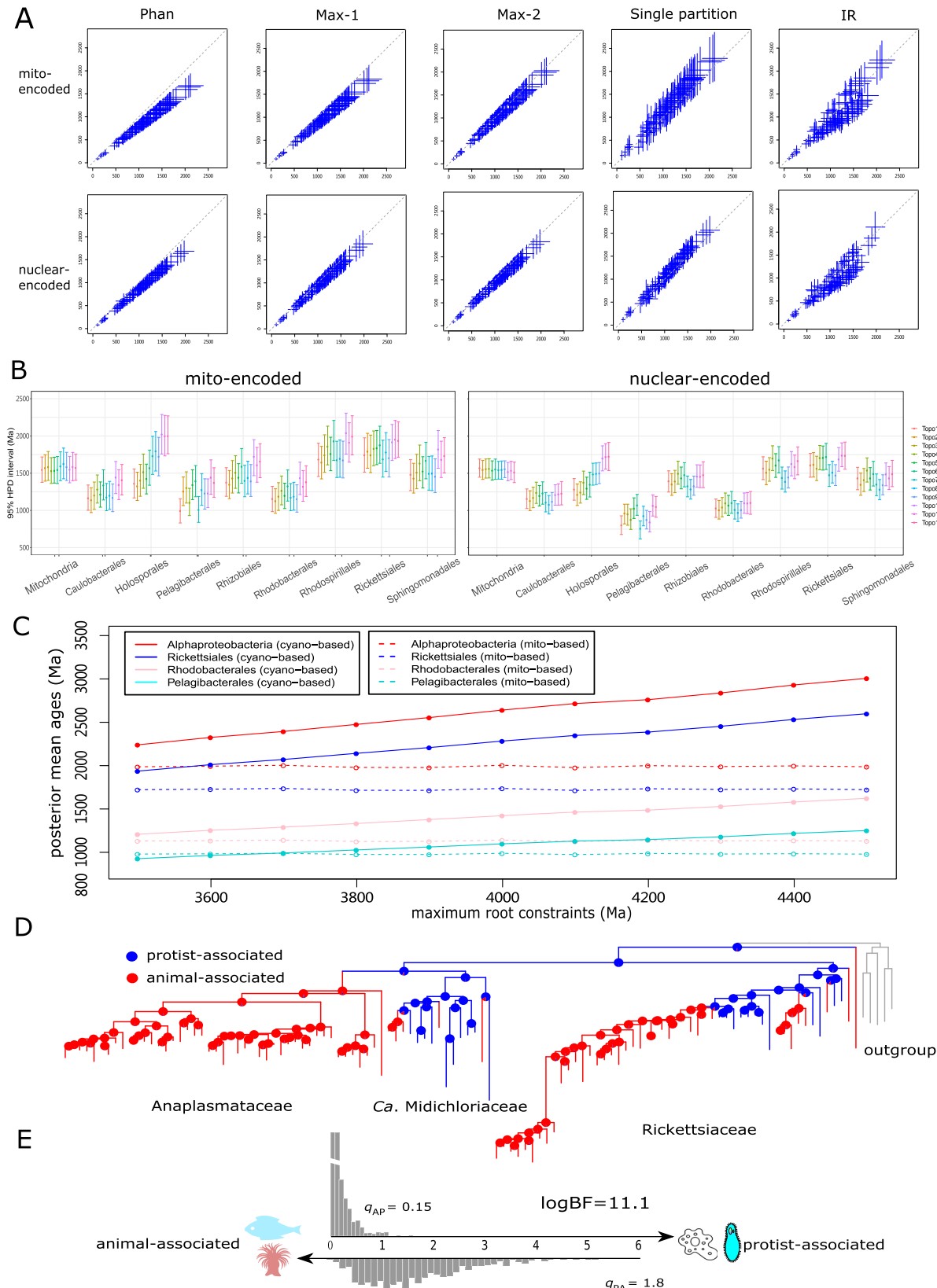

we successively increased the maximum time constraint of the root (i.e., the LCA of *Cyanobacteria* and *Proteobacteria*) from 3500 Ma[41] to 4500 Ma[29], and applied three internal calibration points each of which had only a minimum age constraint (Supplementary Fig. 2B; Supplementary Note 2.2). The results showed

a linear increase in the posterior mean ages of alphaproteobacterial lineages (Fig. 2C). There was an increase of ~30% in the posterior ages if the root maximum age was increased from 3500 Ma to 4500 Ma (Supplementary Fig. 2C). We further applied seven combinations of alternative calibrations and obtained

**Fig. 2 Comparison of the estimated times with alternative dating strategies and ancestral lifestyle reconstruction of *Rickettsiales*. A** The divergence times estimated using alternative schemes (*y* axis; see Supplementary Data 2) versus using the best-practiced scheme (*x* axis) for the mitochondria-based strategy. The best-practiced scheme used a full partition and autocorrelated rates clock model (Supplementary Data 2). The bars in blue indicate the 95% HPDs. *Phan*: only calibration points with Phanerozoic fossils considered; *Max-1*: maximum constraints for nodes with controversial maximum ages removed; *Max-2*: maximum constraints for nodes whose maxima are set as 1891 Ma based on the earliest eukaryotic fossils removed; *Single partition*: all sequences considered as a single partition; *IR*: independent rates clock model (Supplementary Data 2). **B** Changes in the estimated times (95% HPD interval) that result from using different species tree topologies (Supplementary Fig. 1C) using the mitochondria-based strategy. The center corresponds to the posterior mean age. The detailed posterior dates for each clade are shown in Supplementary Table 2. **C** The plot showing the posterior mean ages of the four selected clades estimated under root calibrations from 3500 to 4500 Ma for both the mitochondria-based strategy (mito-encoded data set; dashed line) and the cyanobacteria-based strategy (solid line). The internal calibrations are the same as the ones used as the best-practiced dating scheme for both strategies (Supplementary Data 2). **D** Inferred ancestral hosts of *Rickettsiales*. The pie charts on the nodes show the estimated probabilities of the hosts, and the branch colors indicate the hosts with the higher probability at the corresponding node. Tips represent the randomly selected representative of each OTU (defined by 97% identity of 16 S rRNA gene). **E** The transition rates from animal-associated to protist-associated ($q_{AP}$) and from protist-associated to animal-associated ($q_{PA}$) estimated by the MCMC method in BayesTraits multistate. The log-transformed Bayes factor (logBF) is indicated, where values above 10 are considered very strong evidence for support[51]. The image of ciliates is credited to artist Michael Frey under CC BY-SA 3.0 (https://creativecommons.org/licenses/by-sa/3.0/). No change to the image is made. All other organism images are distributed under CC0.

similar patterns (Supplementary Fig. 12A). In contrast, for the mitochondria-based strategy, using different root maximum constraints obtained similar posterior ages (Fig. 2C; *Root-1*, *-2*, and *-3* in Supplementary Fig. 6). The two strategies showed broadly similar time estimates if the root maximum age in the cyanobacteria-based method was set as 3500 Ma (Fig. 2C) where cyanobacteria and *Proteobacteria* were estimated to split roughly 3400 Mya (Supplementary Fig. 12), a time consistent with some previous estimates[15,29]. This hints that 3500 Ma or a value nearby might be a "reasonable" maximum time constraint for the root when using the cyanobacteria-based method to date the evolution of *Alphaproteobacteria* and potentially other related lineages.

## Discussion

In the present study, we successfully estimated an evolutionary timescale of *Alphaproteobacteria* based on the mitochondrial endosymbiosis, which exhibited higher robustness to the change in the root calibration compared with the traditional cyanobacteria-based approach. Apparently, using cyanobacteria fossils to date *Alphaproteobacteria* evolution heavily relies on root calibration, which is hard to justify. For cyanobacteria, the different maximum time constraints on the root used in different studies partly come from the uncertainty in its phylogenetic position relative to other bacteria: cyanobacteria were inferred to be one of the closest clades to *Proteobacteria*[42], to form an independent clade called Terrabacteria with gram-positive bacteria[15], or to be one of the earliest-split lineages in the bacterial tree of life[43] in different studies. A similar pattern was reported in molecular dating analysis for other prokaryotic lineages[40]. In contrast, the mitochondria-based approach takes the advantages of the abundant phylogenetic, paleontological, stratigraphic, and geochronological data in eukaryotes to assign minimum and maximum bounds to multiple clades, thereby relaxing the dependence of an arbitrarily set root calibration. Specifically, the maximum bound can be achieved based on the youngest geological formation or stratigraphic range that ought to contain fossils of the clade of interest but that does not[44,45]. Indeed, the posterior ages obtained by the mitochondria-based method increased by up to 20% after the removal of maximum constraints of all calibration points except for the root (no max in Supplementary Fig. 5), indicating that the maximum ages imposed by mitochondrial lineages played an important role in constraining the ages of alphaproteobacterial lineages. Note that some prior studies assigned maximum time constraints to calibration points within the cyanobacteria[46,47]. However, these settings are generally based on indirect or contentious evidence,

which potentially leads to posterior ages that might be overly precise (Supplementary Note 2.2)[29,48].

By co-estimating the divergence times of both bacteria and their eukaryotic hosts, it allows directly analyzing their co-evolution with the same data set on the same tree, thereby avoiding methodological artefacts stemming from comparing age estimates from different studies. In addition, the mitochondria-based approach to date evolution is conceptually distinct from the HGT-based method[17,18], as mitochondrial endosymbiosis possibly involves the transfer of thousands of genes and is far more complex than simply an HGT event[19,25]. Consequently, thanks to the many high-confidence orthologs shared by mitochondrial and alphaproteobacterial lineages, our multi-gene data mitigate the challenges faced by Shih et al.[49], which pioneered the idea of endosymbiosis-based dating (see Supplementary Note 3.1). However, our tentative analysis is subject to other challenges such as the lack of internal fossils within the *Alphaproteobacteria* and violation of the molecular clock caused by the fast-evolving nature of mitochondrial genomes (Supplementary Figs. 11, 13). The phylogenetic uncertainties associated with *Holosporales* and *Pelagibacterales*, and unsampled or even extinct lineages could also affect age estimates (Fig. 2B). In addition, the lack of lineages representing the ancestral mitochondrion might generate further uncertainties to molecular clock analysis. Hence, it is always recommended to remember that the estimated divergence times should be interpreted as a span of the posterior age estimate represented by the 95% HPD interval, instead of a time point. New methods and models that better address these issues are necessary to improve the time estimation. It is worth noting that a reduced set of orthologs may be shared by mitochondria and more distantly related bacterial lineages, which limits the application of the mitochondria-based strategy. In this regard, the cyanobacteria-based method is still a powerful method for dating the evolution of non-*Proteobacteria* lineages. Perhaps, it is a good idea to combine mitochondria-based and cyanobacteria-based strategies to date the bacterial tree of life in future studies.

The divergence times estimated here agree with some previous studies[15,16], but are much older than estimated in other studies that constrained the origin time of symbiotic bacteria to be the same as that of their dominant modern hosts, thus ignoring the possibility of host shifts[5,14,50] (Supplementary Note 3.2). Our results suggest that the origin of *Rickettsiales*, where most extant members adapt to an animal-associated lifestyle, predated animals' emergence for ~700 Ma but coincided with the origin of eukaryotes (Fig. 1). Moreover, ancestral lifestyle reconstruction using BayesTraits[51] with a much broader taxon sampling of *Rickettsiales* suggests the LCA of *Rickettsiales* be symbionts of protists and that

the transition rate from protist-associated to animal-associated lifestyle was more than ten times higher than that of the reverse process (Fig. 2D, E and Supplementary Fig. 14). This strongly challenges the view of the concurrence of *Rickettsiales* and animals[50], and instead suggests frequent host transitions from unicellular eukaryotes to animals during *Rickettsiales* evolution. Presumably, the predatory nature of early eukaryotes like amoebae or ciliates might have helped to form a primitive association with ancestral *Rickettsiales*[52]. Specifically, the transmission of protist-harboring *Rickettsiales* to animals likely occurred in aquatic environments where a variety of protists share the same habitats with invertebrates so that animals could have acquired *Rickettsiales* parasites early in their evolution by filter-feeding on infected protists (Supplementary Note 3.3), as hypothesized in other studies[39,53]. This scenario is also supported by recent findings of the association between *Rickettsiales* members with diverse protists[34,54] and by the presence of amoebae-derived genes in some *Rickettsiales* genomes[55]. The common protist hosts of *Rickettsiales* such as amoebae or ciliates feed on bacteria and might harbor stable prokaryotic communities[56,57]. As feedback, protists provide protection against stresses and opportunities of intracellular replication to those able to survive in intra-protist environments. Hence, protists could serve as the training grounds and environmental reservoirs for *Rickettsiales* from which some members of *Rickettsiales* develop invasion mechanisms and find their niches in the animal world to become pathogenic species, as presumably the case for *Legionella*[58,59]. Overall, our results are not contradictory to the idea of host–bacteria co-evolution but suggest more frequent host transitions than previously understood and that caution should be exercised when assigning the divergence time of bacteria based on that of their modern hosts. Besides, the ubiquity of microbial eukaryotes like amoebae in modern environments reminds people to pay more attention to them as potential reservoirs of emerging human diseases.

Although in the present study we focused on host-microbe co-evolution, the mitochondria-based strategy can also be applied to study the biosphere–geosphere co-evolution since the mid-Paleoproterozoic when *Alphaproteobacteria* originated. Further, this strategy may help to calibrate the evolution of related bacteria like *Beta-* and *Gammaproteobacteria*, which requires additional testing and benchmarking. The idea of using endosymbiosis events[46,49] may have a great potential to date the vast majority of the tree of life, where fossil records are lacking.

## Methods

**Phylogenomic reconstruction.** We followed Martijn et al.[21], and used the 24 conserved genes encoded by both the mitochondrial and alphaproteobacterial genomes annotated by MitoCOGs[60] to determine their phylogenetic relationship. Eighty genomes were carefully selected for phylogenomic reconstruction based on prior studies (Supplementary Note 1.1; Supplementary Table 1). Genes were aligned using MAFFT v7.222[61] and trimmed with TrimAl v1.4 ("-st 0.001")[62]. Because *Alphaproteobacteria* is subjected to strong compositional heterogeneity across lineages[21,22,37], which might confound phylogenetic signals and cause phylogenetically unrelated species with similar GC content to cluster together, we recoded the 20 amino acids into four nucleic acid characters according to their physicochemical properties with the dayhoff4 and SR4 recoding scheme, respectively[21,22,37]. Phylogenomic reconstruction was performed under the empirical profile mixture model GTR+G+F+C30 with the PMSF approximation (guide tree GTR+G+I+F) and 1000 ultrafast bootstraps using IQ-Tree v1.6.11[63]. As the trees obtained by dayhoff4 and SR4 recoding schemes showed similar topologies (Supplementary Fig. 1), we used the one obtained by the dayhoff4 recoding scheme for dating (and we subsequently included more eukaryotic taxa as mitochondrial lineages based on the general consensus understanding of the eukaryotic phylogeny for both the mito- and nuclear-encoded data sets in dating [see Supplementary Fig. 2A and Supplementary Note 1.1]).

**Calibration information.** Four calibration points within the Archaeplastida (or Plantae) were selected for the mito-encoded data set, and six additional calibration points from animals, fungi, and rhizarians were included in the nuclear-encoded data set (Supplementary Fig. 2). We based the lower limit of a calibration point upon the most ancient uncontroversial fossil from within the clade. Since fossil records only tell the time that the group of interest had already appeared, the actual origin time of a clade could be more ancient than these minima. Maximum time constraints were determined from the youngest geological formation or stratigraphic range without any members of the clade of interest, as used and recommended by many studies[29,30,64]. Several alternative time constraints were also considered to accommodate the uncertainties in calibration. The full details of calibrations are given in Supplementary Note 2.

**Divergence time estimation.** We compiled two data sets for the mitochondria-based divergence time estimation of *Alphaproteobacteria*, the mito-encoded data set and the nuclear-encoded data set. The mito-encoded set was based on the aforementioned 24 orthologs conserved in *Alphaproteobacteria* and mitochondrial lineages. For the nuclear-encoded set, we retrieved the 29 genes that are likely transferred from the mitochondrial genome to the nuclear genome during the early evolution of eukaryotes from Wang and Wu[3], and excluded seven genes involving putative paralogs or HGTs (Supplementary Data 1). Wrongly annotated sequences in each alignment were removed by manually checking the alignment and gene phylogeny. For the outgroup, we used the two genomes of *Magnetococcales*, the lineage most closely related to *Alphaproteobacteria* and mitochondria, as more distantly related lineages from *Beta-* and *Gammaproteobacteria* share fewer orthologs with *Alphaproteobacteria*.

To alleviate the impacts of mutational saturation, we used amino acids in the main analysis but also repeated the analysis with nucleotide sequences (only the first two codon positions). Dating analyses were predominantly carried out with the approximate likelihood calculation with MCMCTree 4.9j[65], and we also examined the consistency of the results with PhyloBayes v4.1b[66]. A constraint tree constructed using the dayhoff4 recoding scheme described above was applied as both MCMCTree and PhyloBayes require a fixed phylogeny topology. Because previous studies often came to different topologies of the *Alphaproteobacteria* phylogeny, which are mainly associated with the positions of mitochondria, *Holosporales* and *Pelagibacterales*[20–22,37], we considered two, two, and three distinct topologies for these three orders respectively, totaling $2 \times 2 \times 3 = 12$ topologies (Supplementary Fig. 1C; Supplementary Note 1.2). We addressed the phylogenetic uncertainty by repeating MCMCTree analysis with each alternative tree topology. We further selected a best-practiced dating scheme by investigating the impact of the calibration information, clock model (Supplementary Note 2.4, Supplementary Data 3), number of partitions, and cross-lineage rate variation on the estimated posterior ages (see Supplementary Note 1.2 for details). We also used BEAST v2.6.3[67] to perform a co-estimation of the tree topology and divergence time (Supplementary Note 1.2).

The burn-in, sampling frequency, and the number of the iterations were adjusted to 200,000, 100, and 20,000, respectively, based on the results of testing runs. This ensured that the effective sample size for the vast majority of parameters were above 200, as commonly recommended for MCMC-based Bayesian phylogenetic inference[68]. Convergence was assessed by comparing the posterior means from two independent chains and with Tracer v1.6 (http://tree.bio.ed.ac.uk/software/tracer/). The posterior ages were compared with effective priors ("usedata = 0") to ensure that their distributions were different and thereby the sequences used in MCMCTree analysis were informative (Supplementary Table 2). Further, we followed the above procedure to date the divergence time of *Alphaproteobacteria* using the traditional strategy where all calibration points were placed within the cyanobacteria for comparison (see Supplementary Note 2.2).

**Reporting summary.** Further information on research design is available in the Nature Research Reporting Summary linked to this article.

## Data availability

All of the sequences, phylogenetic trees and molecular dating analysis results are available at FigShare https://doi.org/10.6084/m9.figshare.12763547. Specifically, the mito- and nuclear-encoded data sets are provided under the folders dating-aa/mito-encoded and dating-aa/nuclear-encoded in the above online repository, respectively. Raw sequence data are retrieved from the following web-links: https://www.ncbi.nlm.nih.gov/genbank, ftp://ftps.ncbi.nih.gov/pub/koonin/MitoCOGs, https://ensemblgenomes.org, https://bioinformatics.psb.ugent.be/plaza, https://www.uniprot.org. The genome sources are also listed in Supplementary Table 1.

## Code availability

The Ruby[69] codes used to analyze the data are available at https://doi.org/10.6084/m9.figshare.12763547.

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

## Acknowledgements

We are particularly grateful to Jan Janouškovec from University of Oslo for his insights on endosymbiosis and constructive comments on the draft of the manuscript. We thank Mario dos Reis from QMUL, Richard Brown from LJMU, Charles Foster from USYD, and Joana Wolfe from MIT for guidance in molecular dating, and Sergio Muñoz-Gómez from Dalhousie University for help in phylogenomics analysis. We thank lab members Tianhua Liao and Hao Zhang for the discussion, and Kwok Chu Cheung for data retrieving. This work is supported by the National Key R&D Program of China (2018YFC0309800), the Shenzhen Science and Technology Committee (JCYJ20180508161811899), the Hong Kong Branch of Southern Marine Science and Engineering Guangdong Laboratory (Guangzhou) (SMSEGL20SC02), the Hong Kong Research Grants Council Area of Excellence Scheme (AoE/M-403/16), the Direct Grant of CUHK (4053495), and The CUHK Impact Postdoctoral Fellowship Scheme (to S.W.).

## Author contributions

S.W. and H.L. conceived the study, analyzed the data, and wrote the manuscript. S.W. performed the analysis.

## Competing interests

The authors declare no competing interests.
