## [Peer Review File · Nature Communications]

Reviewers' Comments:

Reviewer #1:

Remarks to the Author:

In this manuscript, the authors perform a detailed molecular clock analysis of Alphaproteobacteria, using direct fossil calibration via application of eukaryotic fossil constraints to mitochondrial lineages included within the Alphaproteobacterial tree. The authors show that these results are consistent with and more precise than bacterial molecular clocks constrained by the use of cyanobacterial fossil calibrations. The authors show Proterozoic diversification of major Alphaproteobacterial orders, with a Paleoproterozoic age for the origin of the Alphaproteobacterial class. The authors further show that obligate intracellular bacteria of the order Rickettsiales likely diversified much earlier than their animal host lineages, suggesting a deeper history of co-evolution with eukaryotic hosts.

This work has been very carefully done:

(1) The authors have fairly and carefully accounted for uncertainty in Alphaproteobacterial phylogenetic reconstruction by evaluating divergence times using alternative phylogenies for Alphaproteobacterial orders. They show that these results are robust to these uncertainties, which is expected given the short branches involved in these deep splits, and the lack of the interpretation of any constraints being dependent upon any specific relationship between these orders. The authors have done the same for alternative phylogenies of crown group Eukaryotes.

(2) The authors have accounted for the compositional bias and rate heterogeneity across Alphaproteobacterial groups, by testing phylogenies with Dayhoff recoding of alignments, and testing both uncorrelated and autocorrelated branch rate models;

(3) The authors have fairly tested the impact of using mitochondrial protein sequences encoded in both the mitochondrial and nuclear genomes, in order to account for the impact of the differential evolutionary rates experienced by these genes within different genome settings.

(4) The authors have carefully documented and applied eukaryotic fossil calibrations to the mitochondrial lineages, consistent, as far as I can tell, with previously published work in eukaryote fossil calibration studies. The authors have also accounted for ambiguity in this record (e.g., allowing for Bangiomorpha to only be a minimum age bound for total group red algae).

The methodology and results and supporting information are carefully and clearly documented, and the conclusions are generally strongly supported by the work. The results are useful and novel, with the potential to be applied to many future studies of microbial evolution that depend upon divergence time estimates for major groups of alphaproteobacteria, including soil microbes, marine microbes important to global biogeochemical cycles, and eukaryote-host symbioses. As such, they are relevant and will be of interest to many across a range of scientific disciplines.

The authors make a point of comparing these divergence time estimates to an approach using only cyanobacterial fossil calibrations, as previous studies have done. Since these fossil constraints are very far from the Alphaproteobacteria in the bacterial tree of life, it is unsurprising that they only weakly impact ages within Proteobacteria (although performing the comparison is a useful exercise). While this part of the study is not essential to the key results of the work, it does contain the most problems with respect to fossil calibration.

MAJOR COMMENTS:

There are two major concerns that should be addressed in any revision of the manuscript, as described below:

(1a) Proteobacteria Root Priors. The authors use multiple approaches to establishing root priors for their molecular clock analyses. For the mitochondrial analysis, a root prior for the divergence of Alphas from the Beta/Gamma group was obtained both by integrating across previously estimated posterior age ranges, and "alternatively" by taking generally uninformative, broad uniform priors. The former approach should not be used, as it over-specifies the prior using information used to calculate the posterior in other analyses. Much of this information is similar to the information being added in the posterior age estimates of the author's analysis. In practice, this may not impact the conclusions much, but the results using the "alternative" priors are more sound and appropriate, and these should be the ones reported.

(1b) Cyanobacteria/Proteobacteria root priors. The root priors placed on the Cyano/Alpha common ancestor (root) node are problematic. This node is congruent, or close to congruent, with the Last Bacterial Common Ancestor (LBCA), even though the intervening bacterial groups are not included within the authors' tree (which is fine). As such, this is an extremely ancient divergence. Admittedly, it is hard to identify "reasonable" root priors for such ancient groups without unfairly biasing towards some particular interpretations. The authors choose a root prior with a minimum age of 2320 Ma, and a maximum age of 3500 Ma. The younger-bound on the prior was chosen due to the GOE, which must have followed the origin of oxygenic photosynthesis. However, the problem with this inference is that, it includes as part of the prior hypotheses in which the age of the LBCA pushes right up against the GOE; this does not account for any of the deep bacterial diversifications leading up to the divergence of the cyanobacterial ancestor from its non-photosynthetic relatives (see Soo et al., 2017). Similarly, the older-bound on the prior is far too young, effectively excluding any age for the LCBA older than the first observed well-preserved bacterial fossils. While older-bounds based on the absence of fossil horizons are suitable for groups with a rich and abundant fossil record across strata with high preservation potential, given the sparseness of the archaeal record, and uncertain preservation environments, it is likely that the age of the origin of major bacterial groups greatly precede their first appearance in the fossil record (see Marshall 2019, <https://www.ncbi.nlm.nih.gov/pmc/articles/PMC6871265/>). Subsequently, the earliest bacterial fossil evidence at 3500 Ma would, in actuality, serve as a good YOUNGER bound on the Cyanobacteria/Alphaproteobacteria split, as the LCBA likely precedes these preserved representatives, especially those represented by stromatolite formations at ~3500 Ma. The "alternative" older bound of 3800 Ma referenced by the authors is far more reasonable, and may even be overly conservative, as it assumes a narrative wherein habitability was unlikely in the Hadean (see Abramov & Mojzsis, 2009 for an alternative perspective). As such, the authors should update the cyanobacterial analysis using these more reasonable root priors.

(2a) Cyanobacterial fossil calibrations. The same critique from (1b) applies here as well. The authors carefully cite key fossil calibrations that have been used in previous molecular clock studies. However, there are two problems with the implementation (which may also be problems with how these fossils were used in previous studies as well). First, using first occurrence of specific fossil forms as older bounds (e.g., Pleurocapsales <1900 Ma) assumes that the lack of older similar forms in the fossil record is evidence that the clade containing the character has not yet evolved, rather than (1) lack of preservational environment; (2) low abundance; or (3) divergence of the clade before acquisition of the preserved character. (once again, see Marshall 2019). This application results in likely false precision at critical nodes (e.g., in this case, 1700-1900 Ma), which can propagate to false precision in age estimates across the tree. The maximum age inferred for Nostocales uses a different kind of inference (the physiological constraint of O₂ limitation), which, while it can be debated, is a reasonable assumption independent of the bias in the fossil record.

(2b) stem vs. crown constraints. As described, the cyanobacterial fossil calibrations used should be placed on the total group nodes for the specific bacterial clades, rather than the crown age. For example, if heterocysts are inferred to be shared characters for Nostococales, and thus constrain the age of this group, this calibration should be placed on the ancestor of stem Nostococales, rather than the crown node, since the character arose in the stem and was inherited by extant members of the crown.

Placing the calibration on the crown node (as has apparently been done here, given the authors description in the SI), precludes the possibility that the fossil represents a stem lineage diverging before the crown group, but after the preserved character(s) have evolved. The cyanobacterial molecular clock estimates should therefore be revised to reflect total group, rather than crown group age constraints. Similarly, the eukaryotic fossil calibrations should be carefully evaluated and compared to the literature to ensure that these calibrations were properly placed as well.

If revised molecular clock estimates, as described above, substantially impact posterior age estimates for alphaproteobacteria, the results and conclusions of the manuscript should be revised to reflect this.

Reviewer #2:

Remarks to the Author:

In their manuscript, Wang and Luo apply molecular dating approaches to ascertain when Alphaproteobacteria first diverged from other bacteria using as calibration points well-dated fossils from the eukaryotic fossil record. The use of this strategy is made possible by the fact that eukaryotic mitochondria evolved from Alphaproteobacteria. While the approach is original, it is also very risky given that mitochondrial genes evolve fast and that the closest known alphaproteobacterial relatives of mitochondria seem not to be yet known, which might result in problems of phylogenetic reconstruction and dating. The authors have carried out quite extensive analyses trying to diminish such problems. However, some of the premises and assumptions taken are questionable and the writing of the manuscript, relatively conceptually poor.

Major concerns:

- My most important comment has to do with the a priori choice to use 2530 +/- 153 Ma as divergence time for the 'root', that is the split of Alphaproteobacteria from Beta/Gammaproteobacteria. The present work is conceptually justified by the lack of good calibration points for prokaryotes, the occurrence of Cyanobacteria by the time of the Great Oxidation Event ~2.4 Ga ago being the only real solid calibration point for bacteria. To overcome this problem, the authors aim at using calibration points from well-dated eukaryotic fossils (in more apical branches). However, they also include this 'root' calibration point to set up a deep boundary age for the divergence of Alphaproteobacteria. This completely undermines the objective of their work, since this date has been secondarily estimated from other studies that failed to have good calibration points for prokaryotes. Furthermore, the authors discard some dates that they consider, with reason, to be affected by artifacts and that would place this split at only 606 Ma ago. But the other estimates that they use (Supplementary Note S2) may be as poor. The choice of this date (with, on top, a ridiculously small time error, only 153 Ma) seems excessively subjective and contradictory with the philosophy of their approach. Wang and Luo somehow recognize this, since they also tried alternative prior distributions, including intervals (also subjectively chosen) of 3000 to 1500, 2000 or 1000 Ma. This is explained very briefly in Supplementary Note S2 but not at all discussed in the text. Only a brief mention to "more conservative estimates providing older ages" is presented in the main manuscript. The authors should explain what "more conservative" means, and discuss the implications of these results. They should also try calibration without setting any old boundary date for the Alphaproteobacteria divergence at all. Do they get congruent results?

- I have another concern with the fact that the alphaproteobacterial lineage from which mitochondria derive has not yet been identified. Or else, it is known but a long-branch attraction artifact moves mitochondria to the base of the alphaproteobacterial tree. The authors play with different conditions and datasets to limit potential LBA, but it is not guaranteed that LBA is not affecting the trees. Can the authors use AU-tests or similar to show that other topologies can be statistically excluded? How do the authors think that the inclusion of a new group closer to mitochondria might affect the dating? What

would happen if mitochondria is related to already known alphaproteobacterial clades such as the Rickettsiales? The mitochondria-Rickettsia relationship was favored in the past. Even if compositional bias could affect that result, if Rickettsia constitute a really old clade, there might be some real connection. Can the authors exclude it? These elements and how they would affect dating should be more directly treated in the manuscript; they are not. The discussion is limited.

- While technical aspects of this work seem well developed (the authors have carried out multiple analyses aiming to control for heterogeneous evolutionary rates and taken into account two different datasets), the manuscript writing is conceptually poor in several evolutionary aspects, with some naive or non-accurately expressed ideas, and not necessarily well informed literature-wise. There are many such examples throughout the text. Some of them follow

- The title is inappropriate. Alphaproteobacteria have been evolving since they diverged and continue to evolve. The authors refer to the divergence of extant known Alphaproteobacterial lineages.

- Line 22, abstract, they allude to the "endosymbiosis theory" according to which mitochondria evolved from Alphaproteobacteria. This is not a theory. Neither in popular sense (=hypothesis; since the endosymbiotic origin of mitochondria from alphaproteobacteria is a historical event documented by molecular phylogenetic analyses and not something hypothetical), nor in scientific sense (such as e.g. the theory of relativity; deterministic principles that can be demonstrated mathematically).

- Lines 64-66. 'Some studies assumed a strict relationship of bacteria-host evolution, and calibrated the evolution of pathogenic/symbiotic bacteria based on the divergence time of their modern hosts (mostly animals and plants)'. Which studies? Appropriate references are missing. Please, revise the references provided in general and their adequacy.

- Line 73, "if the donor of an HGT event does not have fossil records while the recipient does". Please, rephrase for accuracy.

- Lines 77-78, 'Alphaproteobacteria based on the endosymbiosis theory that the mitochondrion was transferred to a host nucleus from a bacterial lineage'. Wang and Luo wrongly cite Margulis for this assertion. Margulis talked about the 'nucleocytoplasm' to differentiate the host cell from bacteria-derived organelles. She never said that the alphaproteobacterium was transferred to the host nucleus. To my knowledge, nobody has ever proposed such a hypothesis (nuclear mitochondria), since mitochondria sit in the cytoplasm, not in the nucleus.

- Line 170, 'the topology of Alphaproteobacteria'. The authors do not refer to the 3D structure of Alphaproteobacteria, it seems to me. Rephrase.

Reviewer #3:

Remarks to the Author:

The manuscript aims at providing a reliable dating of different alphaproteobacterial clades leveraging the eukaryotic fossil record. The authors argue that given the importance of alphaproteobacteria in symbiotic relationships with eukaryotes, an accurate timeline is a valuable tool to better understand their evolution as a phylum and in host-symbiont relations.

Overall, the study is well-constructed and addresses most of the common sources of uncertainty in molecular dating (e.g., calibration boundaries, phylogenetic inaccuracies). It reaches a timeline that is similar to some previous studies and makes a convincing argument as to why it differs from other studies. However, one of the major points raised early in this manuscript is that the use of the mitochondrial symbiotic event is a more reliable source of information for calibrating the timetree of alphaproteobacteria compared to others (cyanobacterial) sources of calibrations. It was a bit disappointing to see that this point was not directly addressed in the results/discussion but rather left

to the supplementary material (and even here it is only partially addressed). Discussing this point is especially important because it speaks to the novelty of this study.

First, the argument of the authors in favor of using eukaryotic vs. cyanobacterial calibrations is the large evolutionary distance (and therefore rate variation) between alphas and cyanos compared to alphas and eukaryotes. This is only partially correct because it has been shown that Bacteria and Eukaryotes evolve at different rates, which could nullify the advantage of having a shorter evolutionary distance. Indeed, panels B and C of Fig. S10 show that, for the AR model, only the rate comparisons between Rickettsiales and mitochondria are non-significant while all other comparisons have significantly different rates. Even though relaxed molecular clocks are expected to account for rate variation, the data shown by the authors does not allow a direct evaluation of the effect of this rate variation. One possible way to address this, would be to show similar box plots for the comparisons of rates between alphas and cyanos to determine if the rate variation is larger than that observed between alphas (the non-Rickettsiales) and eukaryotes. If it is, the mitochondria-based method suggested by the authors would be supported.

Second, Fig. S9 shows the comparison of node times for the cyano and the mitochondria-based methods. This figure is difficult to compare because the y-axis scale differs but the time estimates do not appear to be dramatically different. A more useful figure, perhaps, would be a scatter plot with mitochondria vs. cyano-based time estimates as the slope of the regression would provide a good measure of under or over-estimation. In any case, given that the node ages do not seem dramatically different, why is the mitochondria-based method argued to be better? A similar argument is based also on the infinite-plot slopes of Fig. 1 for the nuclear-encoded vs. fig. 2C for the cyano-based times. Also in this case though, the observed difference between the two (0.214 and 0.258 respectively) is comparable to the difference observed between mitochondrial and nuclear-based time estimates. Thus, also based on this evidence, it is unclear why the mitochondria-based method would be more appropriate than the cyanobacteria-based one.

Based on these observations, it seems that the premise of the authors (that mitochondria-based time estimates are more reliable than cyanobacteria-based ones) may not be true (at least not significantly so). This does not invalidate the results of the authors but it questions the novelty of the study that, right now, is focused on the methodology chosen for calibrations. Other important points that make this study significant, such as the frequent host-shifts between protists and animals, are not prominent in this manuscript but might be a more novel result that the timeline itself.

Minor comments:

1. I would prefer to see the boundaries of the calibrations used in the main text (retaining their explanations in the supplementary) as these are key to the results being discussed.
2. The alignment length for the two datasets used should also be listed in the main text.
3. I think the legend and the panels in Fig. S5 do not match (it looks like the panels are identical to Fig. S6)
4. Minor typos are present in the manuscript that should be corrected.
5. The actual distribution used in MCMCTree for the calibrations should be provided (especially in the case of cyanobacteria, it is not mentioned if the boundaries used were soft and what distribution was used; the authors mention uniform but do they mean Cauchy? and if so, what were the tails set at? the default is 0.25 on each side, if I recall correctly).

→We appreciate the editor and three reviewers for their valuable time and valuable comments, which have greatly helped us improve the manuscript. We have carefully considered and accommodated these suggestions. We provide point-to-point responses as follows. Please note that the line numbers correspond to the clean version of the resubmitted manuscript.

REVIEWER COMMENTS

Reviewer #1 (Remarks to the Author):

In this manuscript, the authors perform a detailed molecular clock analysis of Alphaproteobacteria, using direct fossil calibration via application of eukaryotic fossil constraints to mitochondrial lineages included within the Alphaproteobacterial tree. The authors show that these results are consistent with and more precise than bacterial molecular clocks constrained by the use of cyanobacterial fossil calibrations. The authors show Proterozoic diversification of major Alphaproteobacterial orders, with a Paleoproterozoic age for the origin of the Alphaproteobacterial class. The authors further show that obligate intracellular bacteria of the order Rickettsiales likely diversified much earlier than their animal host lineages, suggesting a deeper history of co-evolution with eukaryotic hosts.

This work has been very carefully done:

(1) The authors have fairly and carefully accounted for uncertainty in Alphaproteobacterial phylogenetic reconstruction by evaluating divergence times using alternative phylogenies for Alphaproteobacterial orders. They show that these results are robust to these uncertainties, which is expected given the short branches involved in these deep splits, and the lack of the interpretation of any constraints being dependent upon any specific relationship between these orders. The authors have done the same for alternative phylogenies of crown group Eukaryotes.

(2) The authors have accounted for the compositional bias and rate heterogeneity across Alphaproteobacterial groups, by testing phylogenies with Dayhoff recoding of alignments, and testing both uncorrelated and autocorrelated branch rate models;

(3) The authors have fairly tested the impact of using mitochondrial protein sequences encoded in both the mitochondrial and nuclear genomes, in order to account for the impact of the differential evolutionary rates experienced by these genes within different genome settings.

(4) The authors have carefully documented and applied eukaryotic fossil calibrations to the mitochondrial lineages, consistent, as far as I can tell, with previously published work in eukaryote fossil calibration studies. The authors have also accounted for ambiguity in this record (e.g., allowing for Bangiomorpha to only be a minimum age bound for total group red algae).

The methodology and results and supporting information are carefully and clearly documented, and the conclusions are generally strongly supported by the work. The results are useful and novel, with the potential to be applied to many future studies of microbial evolution that depend upon divergence time estimates for major groups of alphaproteobacteria, including soil microbes, marine microbes important to global biogeochemical cycles, and eukaryote-host symbioses. As such, they are relevant and will be of interest to many across a range of scientific disciplines.

→We thank the reviewer for the positive comments.

The authors make a point of comparing these divergence time estimates to an approach using only cyanobacterial fossil calibrations, as previous studies have done. Since these fossil constraints are very far from the Alphaproteobacteria in the bacterial tree of life, it is unsurprising that they only weakly impact ages within Proteobacteria

(although performing the comparison is a useful exercise). While this part of the study is not essential to the key results of the work, it does contain the most problems with respect to fossil calibration.

→We thank the reviewer for bringing up the issue related to the cyano-based dating strategy. We have thoroughly revised this part of analysis and provided our responses as follows.

MAJOR COMMENTS:

There are two major concerns that should be addressed in any revision of the manuscript, as described below:

(1a) Proteobacteria Root Priors. The authors use multiple approaches to establishing root priors for their molecular clock analyses. For the mitochondrial analysis, a root prior for the divergence of Alphas from the Beta/Gamma group was obtained both by integrating across previously estimated posterior age ranges, and "alternatively" by taking generally uninformative, broad uniform priors. The former approach should not be used, as it over-specifies the prior using information used to calculate the posterior in other analyses. Much of this information is similar to the information being added in the posterior age estimates of the author's analysis. In practice, this may not impact the conclusions much, but the results using the "alternative" priors are more sound and appropriate, and these should be the ones reported.

→Our original idea was to take the advantage of the several previous estimates of the divergence times of *Alpha-* and *Beta/Gammaproteobacteria*. But as pointed out by the reviewer, we note the issue of the over-specification of the root prior. We have accordingly made the following changes (see also root calibrations in Note S2.1).

- i) We removed the four species from *Beta/Gammaproteobacteria*, as they do not provide calibration points but the large evolutionary distances between these outgroup species and *Alphaproteobacteria* may bring further uncertainties in dating (L366-L369, line numbers correspond to the clean version of the

manuscript).

- ii) We set the minimum time constraint of the root (i.e., the LCA of *Alphaproteobacteria*/mitochondria and *Magnetococcales*) to 1000 Ma, roughly the same as the crown group of eukaryotes (1033 Ma), based on the fossils of *Bangiomorpha* (L613). Note that eukaryotic fossils of an earlier age cannot be used to inform the prior. This is because these fossils could be derived from stem group of eukaryotes (Betts et al., 2018), and it is unknown whether they had mitochondria in particular considering a late-mitochondria acquisition model (Pittis and Gabaldón, 2016).
- iii) We set the maximum bound of the root as 3000 Ma. We also tried alternative maximum bounds (3500 Ma, 4000 Ma and 4500 Ma), and obtained consistent age estimates (Fig. S5). This suggests that 3000 Ma is sufficiently large to accommodate the uncertainties in the divergence times of the *Alphaproteobacteria*.

We accordingly performed MCMCTree analysis with the new root calibrations. In the following figure, we plot the posterior ages obtained with the root age calibrated by a gamma distribution prior, which was used in the original submission (y-axis), and the revised root calibration (x-axis). With a few exceptions, the results are highly consistent.

(1b) Cyanobacteria/Proteobacteria root priors. The root priors placed on the

Cyano/Alpha common ancestor (root) node are problematic. This node is congruent, or close to congruent, with the Last Bacterial Common Ancestor (LBCA), even though the intervening bacterial groups are not included within the authors' tree (which is fine). As such, this is an extremely ancient divergence. Admittedly, it is hard to identify "reasonable" root priors for such ancient groups without unfairly biasing towards some particular interpretations. The authors choose a root prior with a minimum age of 2320 Ma, and a maximum age of 3500 Ma. The younger-bound on the prior was chosen due to the GOE, which must have followed the origin of oxygenic photosynthesis. However, the problem with this inference is that, it includes as part of the prior hypotheses in which the age of the LBCA pushes right up against the GOE; this does not account for any of the deep bacterial diversifications leading up to the divergence of the cyanobacterial ancestor from its non-photosynthetic relatives (see Soo et al., 2017). Similarly, the older-bound on the prior is far too young, effectively excluding any age for the LCBA older than the first observed well-preserved bacterial fossils. While older-bounds based on the absence of fossil horizons are suitable for groups with a rich and abundant fossil record across strata with high preservation potential, given the sparseness of the archaeal record, and uncertain preservation environments, it is likely that the age of the origin of major bacterial groups greatly precede their first appearance in the fossil record (see Marshall 2019, <https://www.ncbi.nlm.nih.gov/pmc/articles/PMC6871265/>). Subsequently, the earliest bacterial fossil evidence at 3500 Ma would, in actuality, serve as a good YOUNGER bound on the Cyanobacteria/Alphaproteobacteria split, as the LCBA likely precedes these preserved representatives, especially those represented by stromatolite formations at ~3500 Ma. The "alternative" older bound of 3800 Ma referenced by the authors is far more reasonable, and may even be overly conservative, as it assumes a narrative wherein habitability was unlikely in the Hadean (see Abramov & Mojzsis, 2009 for an alternative perspective). As such, the authors should update the cyanobacterial analysis using these more reasonable root priors.

→We have noticed that the calibration of the root (the LCA of cyanobacteria and *Alphaproteobacteria*) is inappropriate. In the revised manuscript, we have used 3000 Ma as the minimum bound of the root. This is based on the recent evidence for the appearance of O₂ on Earth dating back to ~3000 Ma (Crowe et al., 2013; Planavsky et al., 2014), as also used in recent studies (Betts et al., 2018; Zhang et al., 2021). We also used 2320 Ma, which is based on the GOE, as an alternative minimum bound (Note S2.2; Dataset S2).

As mentioned by the reviewer, the maximum bound is more difficult to set. Some of the disputes might arise from the phylogenetic position of cyanobacteria, which are placed closely to Proteobacteria in early studies (Battistuzzi et al., 2004; Battistuzzi and Hedges, 2009) but are moved to a very basal position in the bacterial tree of life in recent ones (Hug et al., 2016; Méheust et al., 2019). Apparently, this could result in different assumptions to the maximum age of the root (i.e., the LCA of cyanobacteria and *Alphaproteobacteria*). Due to this uncertainty, we set different maximum constraints on the root by successively increasing the maximum bound from 3500 Ma (the age of Pilbara Supergroup, which provides one of the earliest convincing evidence for life on Earth) (Allwood et al., 2007) to 4500 Ma (the age of Earth).

As we illustrate in L215-L237, the change in the root maximum age greatly affected dating. Specifically, the posterior ages for alphaproteobacterial lineages increased linearly as the root maximum age increased from 3500 Ma to 4500 Ma (Fig. 2C). We further show that this uncertainty in time estimation affects all alphaproteobacterial clades and is not specific to any calibration set using the cyanobacteria-based strategy (Fig. S12). In contrast, for the mitochondria-based strategy, the posterior ages were robust to the changes in the root calibration (Fig. 2C [dashed line]; Fig. S5). We argue that this is because maximum time constraints of certain eukaryotic clades can be set by carefully leveraging the rich phylogenetic, anatomical, stratigraphic and geochronological data in eukaryotes (Benton and Donoghue, 2007). This relaxes the dependence of time estimates on the root maximum age, which, at least for dating bacterial evolution, is often difficult to justify

but has to be set in a to some extent arbitrary way (Szöllósi et al., 2021) (L240-L260).

(2a) Cyanobacterial fossil calibrations. The same critique from (1b) applies here as well. The authors carefully cite key fossil calibrations that have been used in previous molecular clock studies. However, there are two problems with the implementation (which may also be problems with how these fossils were used in previous studies as well). First, using first occurrence of specific fossil forms as older bounds (e.g., Pleurocapsales <1900 Ma) assumes that the lack of older similar forms in the fossil record is evidence that the clade containing the character has not yet evolved, rather than (1) lack of preservational environment; (2) low abundance; or (3) divergence of the clade before acquisition of the preserved character. (once again, see Marshall 2019). This application results in likely false precision at critical nodes (e.g., in this case, 1700-1900 Ma), which can propagate to false precision in age estimates across the tree. The maximum age inferred for Nostocales uses a different kind of inference (the physiological constraint of O₂ limitation), which, while it can be debated, is a reasonable assumption independent of the bias in the fossil record.

→We realize the inappropriateness of setting the soft maximum time constraints for cyanobacteria lineages, which diverged in ancient times and have poor fossil preservation. Accordingly, we have removed the maximum constraints of all corresponding nodes within the cyanobacteria and re-performed all analyses. In other words, only a minimum bound is provided for each within-cyanobacteria calibration point (Note S2.2; Dataset S2). We have provided the rationales, and have emphasized the difficulty in setting the maximum time bound in cyano-based dating analysis in L256-L260 and Note S2.2.

(2b) stem vs. crown constraints. As described, the cyanobacterial fossil calibrations used should be placed on the total group nodes for the specific bacterial clades, rather than the crown age. For example, if heterocysts are inferred to be shared characters for Nostocales, and thus constrain the age of this group, this calibration should be

placed on the ancestor of stem Nostocales, rather than the crown node, since the character arose in the stem and was inherited by extant members of the crown. Placing the calibration on the crown node (as has apparently been done here, given the authors description in the SI), precludes the possibility that the fossil represents a stem lineage diverging before the crown group, but after the preserved character(s) have evolved. The cyanobacterial molecular clock estimates should therefore be revised to reflect total group, rather than crown group age constraints. Similarly, the eukaryotic fossil calibrations should be carefully evaluated and compared to the literature to ensure that these calibrations were properly placed as well.

→We thank the reviewer for pointing out that setting calibrations on the crown nodes is inappropriate. In the revised manuscript, we have moved all of the three cyanobacteria calibrations to the total group of the corresponding clade (Fig. S2B; Note S2.2), and re-performed the analyses. Note that we followed other studies (Shih et al., 2017; Zhang et al., 2021) to include five genomes belonging to non-oxygenic cyanobacteria to assign the calibration to the total group, instead of the crown group, of oxygenic cyanobacteria based on the evidence of the rise of O₂ (Note S1.2.3).

If revised molecular clock estimates, as described above, substantially impact posterior age estimates for alphaproteobacteria, the results and conclusions of the manuscript should be revised to reflect this.

→As we respond above, we have used updated calibrations to estimate the divergence times using the mitochondria- and cyanobacteria-based strategies. Compared with the original manuscript, the time estimates for *Alphaproteobacteria* were very similar using the mitochondria-based method (Question 1a), but showed considerable variation depending on the root calibrations when the cyanobacteria-based approach was used (Question 1b). The corresponding results (L106-L213) and figures (Figs. 2C, S12) have been accordingly revised.

Reviewer #2 (Remarks to the Author):

In their manuscript, Wang and Luo apply molecular dating approaches to ascertain when Alphaproteobacteria first diverged from other bacteria using as calibration points well-dated fossils from the eukaryotic fossil record. The use of this strategy is made possible by the fact that eukaryotic mitochondria evolved from Alphaproteobacteria. While the approach is original, it is also very risky given that mitochondrial genes evolve fast and that the closest known alphaproteobacterial relatives of mitochondria seem not to be yet known, which might result in problems of phylogenetic reconstruction and dating. The authors have carried out quite extensive analyses trying to diminish such problems. However, some of the premises and assumptions taken are questionable and the writing of the manuscript, relatively conceptually poor.

Major concerns:

- My most important comment has to do with the a priori choice to use 2530 +/- 153 Ma as divergence time for the 'root', that is the split of Alphaproteobacteria from Beta/Gammaproteobacteria. The present work is conceptually justified by the lack of good calibration points for prokaryotes, the occurrence of Cyanobacteria by the time of the Great Oxidation Event ~2.4 Ga ago being the only real solid calibration point for bacteria. To overcome this problem, the authors aim at using calibration points from well-dated eukaryotic fossils (in more apical branches). However, they also include this 'root' calibration point to set up a deep boundary age for the divergence of Alphaproteobacteria. This completely undermines the objective of their work, since this date has been secondarily estimated from other studies that failed to have good calibration points for prokaryotes. Furthermore, the authors discard some dates that they consider, with reason, to be affected by artifacts and that would place this split at only 606 Ma ago. But the other estimates that they use (Supplementary Note S2) may

be as poor. The choice of this date (with, on top, a ridiculously small time error, only 153 Ma) seems excessively subjective and contradictory with the philosophy of their approach. Wang and Luo somehow recognize this, since they also tried alternative prior distributions, including intervals (also subjectively chosen) of 3000 to 1500, 2000 or 1000 Ma. This is explained very briefly in Supplementary Note S2 but not at all discussed in the text. Only a brief mention to “more conservative estimates providing older ages” is presented in the main manuscript. The authors should explain what “more conservative” means, and discuss the implications of these results. They should also try calibration without setting any old boundary date for the Alphaproteobacteria divergence at all. Do they get congruent results?

→ We thank the reviewer for asking the question on the root calibration. We note the issue of the over-specification of the root prior, and have made the following changes. We have accordingly applied a conservative calibration on the root based on a uniform distribution from 1000 Ma to 3000 Ma (L612 [line numbers correspond to the clean version of the manuscript]; see also root calibrations in Note S2.1), and have re-performed all analyses with the updated root calibration.

Specifically, the minimum bound was set as 1000 Ma. This is according to the minimum age of the crown group of eukaryotes (1033 Ma, thus roughly 1000 Ma), based on the fossils of *Bangiomorpha* (L622). Note that we cannot use eukaryotic fossils of an earlier age to set the minimum bound. This is because these fossils might be derived from stem group of eukaryotes (Betts et al., 2018), and it is possible that they originated before the appearance of mitochondria especially given a late-mitochondria acquisition model (Pittis and Gabaldón, 2016).

Since a maximum age at the root is required for MCMCTree, we set a conservative maximum bound, which is 3000 Ma. We also tried alternative maximum time constraints of 3500, 4000 and 4500 Ma, respectively. Because all of these root maximum constraints obtained consistent age estimates for alphaproteobacterial lineages (Fig. S5), the results suggest that 3000 Ma is large enough to accommodate the uncertainties in time estimation. Besides, note that we have removed *Beta*- and

Gammaproteobacteria from dating analysis as they are distantly related to and share fewer orthologs with *Alphaproteobacteria*, which may introduce further uncertainties in dating analysis (L366-L369).

Shown in the following figure are the time estimates obtained from the calibrations used in the original manuscript (y-axis) and those estimated by the updated calibrations (x-axis). The results show that the vast majority of the points fall on the $y=x$ line, which indicates congruent time estimates.

- I have another concern with the fact that the alphaproteobacterial lineage from which mitochondria derive has not yet been identified. Or else, it is known but a long-branch attraction artifact moves mitochondria to the base of the alphaproteobacterial tree. The authors play with different conditions and datasets to limit potential LBA, but it is not guaranteed that LBA is not affecting the trees. Can the authors use AU-tests or similar to show that other topologies can be statistically excluded? How do the authors think that the inclusion of a new group closer to mitochondria might affect the dating? What would happen if mitochondria is related to already known alphaproteobacterial clades such as the Rickettsiales? The mitochondria-Rickettsia relationship was favored in the past. Even if compositional bias could affect that result, if Rickettsia constitute a really old clade, there might be some real connection. Can the authors exclude it? These elements and how they would affect dating should be more directly treated in the manuscript; they are not. The discussion is limited.

→We thank the reviewer for the valuable comments and questions. We realize that phylogenetic uncertainty is a very important issue. We have accordingly performed the following analyses to assess the impact of alternative topologies on dating.

- i) In the original submission, we had considered 2, 2, and 3 possible phylogenetic positions for mitochondria, *Holosporales* and *Pelagibacterales*, respectively, totaling $2 \times 2 \times 3 = 12$ different topologies (L376-L382; Fig. S1C), and performed dating analysis based on each of them. Except *Holosporales* and *Pelagibacterales*, most alphaproteobacterial lineages were only slightly affected as to their time estimates (Fig. 2B). This is likely because the different phylogenies of *Alphaproteobacteria* are mainly associated with the above mentioned three deeply-branching clades.
- ii) In the revised manuscript, we have performed topology tests for the aforementioned 12 topologies using five statistical methods, including the AU test as suggested by the reviewer (Table S3). Topology 1 (the best-practiced tree used in dating analysis) and topology 4 were the only two that passed all tests. The remaining 10 topologies were rejected by at least four out of the five tests. We have emphasized that topology 4, where mitochondria form a monophyletic group with *Rickettsiales*, is likely another possible topology of the *Alphaproteobacteria* phylogeny (L180-L183). This does not affect our results, because the time estimates based on topology 4 were congruent with the one used in the main analysis as illustrated above (Fig. 2B).
- iii) We agree that the current knowledge of the taxonomy of *Alphaproteobacteria* is still incomplete, and it is fully possible that there are unsampled or even extinct early-split *Alphaproteobacteria* lineages. To minimize its impact on dating, we expanded the taxon sampling by including 16 additional metagenome-assembled genomes (MAGs) belonging to *Alphaproteobacteria* (Martijn et al., 2018), many of which are early-split lineages (L184-L189). The estimated dates for *Alphaproteobacteria* obtained with this expanded dataset were generally consistent with the dataset without using these MAGs

(Fig. S8).

- iv) Moreover, in the revised manuscript, we employed BEAST to perform a co-estimation of the tree topology and divergence times (L191-L197). In other words, unlike MCMCTree which requires a fixed tree topology, the tree topology is not fixed in BEAST analysis, which might better reflect the phylogenetic uncertainties in dating. The topology of the tree inferred by BEAST (Fig. S9) differed from the one inferred by our phylogenomic analysis. This is not unexpected, because they use different methods to infer the tree. In particular, the mixture model and amino acid recoding used in our phylogenomic analysis, are not available in BEAST. In general, the time estimates were consistent between MCMCTree and BEAST analyses, although for the nuclear-encoded dataset the BEAST analysis estimated younger posterior ages (Fig. S9).
- v) We have clearly stated the importance of considering phylogenetic uncertainty in dating *Alphaproteobacteria* evolution in Discussion (L273-L279).

- While technical aspects of this work seem well developed (the authors have carried out multiple analyses aiming to control for heterogeneous evolutionary rates and taken into account two different datasets), the manuscript writing is conceptually poor in several evolutionary aspects, with some naive or non-accurately expressed ideas, and not necessarily well informed literature-wise. There are many such examples throughout the text. Some of them follow

→ We thank the reviewer for pointing out the places where the writing is unclear and further clarification is needed. We have addressed all of them and have carefully checked for similar issues throughout the manuscript.

- The title is inappropriate. Alphaproteobacteria have been evolving since they diverged and continue to evolve. The authors refer to the divergence of extant known Alphaproteobacterial lineages.

→We understand that the point mentioned by the reviewer. However, we feel that it might be difficult to accommodate it in the title, which is supposed to be concise. Actually, a similar way of entitling a paper is often seen in molecular dating studies (Berney and Pawlowski, 2006; Dohrmann and Wörheide, 2017; Magnabosco et al., 2018; Zimmer et al., 2007). Therefore, we feel that it is fine to keep the original title. We have clarified this point in L274-L275.

- Line 22, abstract, they allude to the “endosymbiosis theory” according to which mitochondria evolved from Alphaproteobacteria. This is not a theory. Neither in popular sense (=hypothesis; since the endosymbiotic origin of mitochondria from alphaproteobacteria is a historical event documented by molecular phylogenetic analyses and not something hypothetical), nor in scientific sense (such as e.g. the theory of relativity; deterministic principles that can be demonstrated mathematically).

→We have rephrased it as “mitochondrial endosymbiosis” (L27, L72, L133, L241).

- Lines 64-66. ‘Some studies assumed a strict relationship of bacteria-host evolution, and calibrated the evolution of pathogenic/symbiotic bacteria based on the divergence time of their modern hosts (mostly animals and plants)’. Which studies? Appropriate references are missing. Please, revise the references provided in general and their adequacy.

→We have added the citations in L58. We have also carefully revised the references in the manuscript.

- Line 73, “if the donor of an HGT event does not have fossil records while the recipient does”. Please, rephrase for accuracy.

→Rephrased (L67-L68).

- Lines 77-78, ‘Alphaproteobacteria based on the endosymbiosis theory that the

mitochondrion was transferred to a host nucleus from a bacterial lineage'. Wang and Luo wrongly cite Margulis for this assertion. Margulis talked about the 'nucleocytoplasm' to differentiate the host cell from bacteria-derived organelles. She never said that the alphaproteobacterium was transferred to the host nucleus. To my knowledge, nobody has ever proposed such a hypothesis (nuclear mitochondria), since mitochondria sit in the cytoplasm, not in the nucleus.

→We have carefully reviewed the literature and revised the corresponding expression (L73).

- Line 170, 'the topology of Alphaproteobacteria'. The authors do not refer to the 3D structure of Alphaproteobacteria, it seems to me. Rephrase.

→Rephrased (L171).

Reviewer #3 (Remarks to the Author):

The manuscript aims at providing a reliable dating of different alphaproteobacterial clades leveraging the eukaryotic fossil record. The authors argue that given the importance of alphaproteobacteria in symbiotic relationships with eukaryotes, an accurate timeline is a valuable tool to better understand their evolution as a phylum and in host-symbiont relations.

Overall, the study is well-constructed and addresses most of the common sources of uncertainty in molecular dating (e.g., calibration boundaries, phylogenetic inaccuracies). It reaches a timeline that is similar to some previous studies and makes a convincing argument as to why it differs from other studies.

→We thank the reviewer for the positive comments.

However, one of the major points raised early in this manuscript is that the use of the mitochondrial symbiotic event is a more reliable source of information for calibrating the timetree of alphaproteobacteria compared to others (cyanobacterial) sources of calibrations. It was a bit disappointing to see that this point was not directly addressed in the results/discussion but rather left to the supplementary material (and even here it is only partially addressed). Discussing this point is especially important because it speaks to the novelty of this study.

→We thank the reviewer for pointing out this issue. We have thoroughly revised this part, added a subsection to address this issue in the main text (L215-L236, line numbers correspond to the clean version of the manuscript), and provided stronger reasons for which we think that the mitochondria-based strategy provides more reliable time estimates (L238-L259) as follows.

First, the argument of the authors in favor of using eukaryotic vs. cyanobacterial calibrations is the large evolutionary distance (and therefore rate variation) between alphas and cyanos compared to alphas and eukaryotes. This is only partially correct

because it has been shown that Bacteria and Eukaryotes evolve at different rates, which could nullify the advantage of having a shorter evolutionary distance. Indeed, panels B and C of Fig. S10 show that, for the AR model, only the rate comparisons between Rickettsiales and mitochondria are non-significant while all other comparisons have significantly different rates. Even though relaxed molecular clocks are expected to account for rate variation, the data shown by the authors does not allow a direct evaluation of the effect of this rate variation. One possible way to address this, would be to show similar box plots for the comparisons of rates between alphas and cyanos to determine if the rate variation is larger than that observed between alphas (the non-Rickettsiales) and eukaryotes. If it is, the mitochondria-based method suggested by the authors would be supported.

→ We have elaborated this point with more detailed analyses (L200-L213). We first estimated the average substitution rate of gene using MCMCTree, and calculated the relative difference in substitution rate, calculated as $\frac{|rate[mito] - rate[\alpha]|}{\max(rate[mito], rate[\alpha])}$, between mitochondria and non-Rickettsiales Alphaproteobacteria. Basically, a lower value of rate relative difference denotes lower among-branch rate heterogeneity (vice versa). Then, we applied different cutoffs of rate relative difference to divide genes into different categories. Finally, we performed MCMCTree analysis to estimate the posterior ages for genes of different rate categories (Fig. S11). The logic is simple: if the relaxed molecular clock model failed to accommodate different substitution rates between mitochondria and Rickettsiales, it should be observed that the time estimates were biased towards rate heterogeneous (or homogeneous) genes.

As shown in Fig. S11 and in L200-L213, there appears no apparent bias and that the 95% HPD interval becomes smaller as more genes are included. Further, allowing larger among-branch rate variation obtained similar time estimates (*sigma* in Fig. S5; L210-L212). Besides, the genes used in mitochondria-based dating are based on Martijn et al., 2018 and Wang and Wu, 2015. These two studies have carefully selected the genes that show less among-branch rate variation for phylogenomic analysis. Last, Fig. S13 indicates that significant rate differences between

mitochondria and *Alphaproteobacteria* were detected only using the AR model, the best-fit model based on mcmc3r analysis. Thus, the rate differences may have already been accounted for in our dating analysis. The two less preferred models (STR and IR) failed to detect the rate differences and led to considerably younger time estimates (Fig. S4). Based on these lines of evidence, we think that the rate variation among branches i) was minimized in our analysis, and ii) did not have a large impact on time estimates. We have also stated the limitations of our approach in L270-L278.

As suggested by the reviewer, we also estimated the substitution rates for cyanobacteria. We did not detect significant differences in rate between cyanobacteria and non-*Rickettsiales Alphaproteobacteria*, as shown in the following figure ($p = 0.41$, paired Wilcoxon test).

However, because the differences in substitution rate between mitochondria and *Alphaproteobacteria* are likely well accounted for by the relaxed molecular clock model (see above), and because as we illustrate below, the cyanobacteria-based strategy faces the problem that its time estimates are too sensitive to the root calibration, we argue that the issue in rate heterogeneity is minimized in our analysis and may not have large impacts on the time estimates.

Second, Fig. S9 shows the comparison of node times for the cyano and the mitochondria-based methods. This figure is difficult to compare because the y-axis scale differs but the time estimates do not appear to be dramatically different. A more useful figure, perhaps, would be a scatter plot with mitochondria vs. cyano-based time estimates as the slope of the regression would provide a good measure of under or over-estimation. In any case, given that the node ages do not seem dramatically different, why is the mitochondria-based method argued to be better? A similar argument is based also on the infinite-plot slopes of Fig. 1 for the nuclear-encoded vs. fig. 2C for the cyano-based times. Also in this case though, the observed difference between the two (0.214 and 0.258 respectively) is comparable to the difference observed between mitochondrial and nuclear-based time estimates. Thus, also based on this evidence, it is unclear why the mitochondria-based method would be more appropriate than the cyanobacteria-based one.

→ We thank the reviewer for raising the important question about the comparison between mitochondria- and cyanobacteria-based strategies. At first, we would like to point out that in the revised manuscript we followed Reviewer 1's suggestions to update the calibrations on cyanobacteria lineages. The major changes are i) the use of alternative calibrations (uniform distribution) on the root (*Alphaproteobacteria*/mitochondria vs. *Cyanobacteria*), and ii) the removal of the maximum time constraints of calibration points within the cyanobacteria. We also realize that it is difficult to determine a reasonable root calibration, hence we accommodated this uncertainty by using different maximum time constraints on the root from 3500 Ma (the earliest convincing evidence for life on Earth) (Allwood et al., 2007) to 4500 Ma (the age of Earth) (Note S2.2).

We have updated all relevant results (L215-L237). The new results show that the posterior estimates of alphaproteobacterial lineages heavily depended on the calibration of the root (*Alphaproteobacteria*/mitochondria vs. cyanobacteria) using the cyano-based method. As shown in Fig. 2C, the estimates of the origin times of four

select clades, namely *Alphaproteobacteria*, *Rickettsiales*, *Rhodobacterales*, and *Pelagibacterales*, increased by ~30% when the maximum time constraint on the root changed from 3500 Ma to 4500 Ma. The same pattern remained after we used seven alternative calibration sets in cyanobacteria (Fig. S12A). In contrast, the time estimates are much less dependent on the root calibration for the mitochondria-based strategy (Figs. 2C, S5). As suggested by this reviewer, we also used scatter plots to compare the posterior estimates between the mitochondria- and cyano-based strategies, and obtained similar patterns, particularly for nodes of an older age (Fig. S12B). The mito- and cyano-based strategies give generally consistent estimates when the root maximum age was set as 3500 Ma for the cyano-based strategy.

The above analyses suggest that the mitochondria-based dating strategy provides more reliable results, as it is much less dependent on the root calibration which is often difficult to justify for dating bacterial evolution. We provide detailed reasons for this finding in L240-L260. For the mitochondria-based method, the abundant fossil records in eukaryotes provide the opportunity to determine reasonable calibrations for certain clades (particularly within animals and plants but also including other lineages), based on a combination of anatomical, stratigraphic and geochronological evidence (Benton and Donoghue, 2007; Donoghue and Benton, 2007). In contrast, due to the poor conservation of fossil records within the cyanobacteria, it is difficult to determine the maximum time constraint for the calibration points within the cyanobacteria (Zhang et al., 2021). As a result, typically an (to some extent) arbitrary maximum age has to be set to the root, which could lead to very different estimates based on different root calibrations.

In addition, we agree that it may be inappropriate to compare the slopes in the infinite-sites plots. This is because the genomes and calibration points are different between the mitochondria- and cyano-based strategies (and do not share any calibration point). We have removed the infinite-sites plot for the cyano-based method, and kept only the plots for the mitochondria-based method in Fig. 1.

Based on these observations, it seems that the premise of the authors (that mitochondria-based time estimates are more reliable than cyanobacteria-based ones) may not be true (at least not significantly so). This does not invalidate the results of the authors but it questions the novelty of the study that, right now, is focused on the methodology chosen for calibrations. Other important points that make this study significant, such as the frequent host-shifts between protists and animals, are not prominent in this manuscript but might be a more novel result than the timeline itself.

→As described above, we have provided the detailed reasons for why we think that the times obtained with the mitochondria-based strategy are more reliable than those estimated by the cyanobacteria-based approach (L240-L260). Briefly, the mitochondria-based strategy displays higher robustness to the changes of the root calibration, which is often set in an (to some extent) arbitrary way. We have also expanded the discussion on host shift and its implications for bacterium-host co-evolution and the prevention of the emergence of human pathogens (L293-L314).

Minor comments:

1. I would prefer to see the boundaries of the calibrations used in the main text (retaining their explanations in the supplementary) as these are key to the results being discussed.

→We have provided this information in L606-L616.

2. The alignment length for the two datasets used should also be listed in the main text.

→Provided (L605).

3. I think the legend and the panels in Fig. S5 do not match (it looks like the panels are identical to Fig. S6

→We are sorry for this mistake. It is corrected in the resubmitted manuscript.

4. Minor typos are present in the manuscript that should be corrected.

→We have carefully checked the grammar and corrected the typos throughout the manuscript.

5. The actual distribution used in MCMCTree for the calibrations should be provided (especially in the case of cyanobacteria, it is not mentioned if the boundaries used were soft and what distribution was used; the authors mention uniform but do they mean Cauchy? and if so, what were the tails set at? the default is 0.25 on each side, if I recall correctly.

→This was mentioned in the Note S1 in the original submission. However, we agree that it is necessary to provide this information in a clearer way. Accordingly, we have described it in L614-L616 in the main text and Note S2. We have also provided the visualization of the distributions of calibration densities in Fig. S2.

References

- Allwood AC, Walter MR, Burch IW, Kamber BS. 2007. 3.43 billion-year-old stromatolite reef from the Pilbara Craton of Western Australia: Ecosystem-scale insights to early life on Earth. *Precambrian Res* **158**:198–227.
doi:10.1016/j.precamres.2007.04.013
- Battistuzzi FU, Feijao A, Hedges SB. 2004. A genomic timescale of prokaryote evolution: Insights into the origin of methanogenesis, phototrophy, and the colonization of land. *BMC Evol Biol* **4**:44. doi:10.1186/1471-2148-4-44
- Battistuzzi FU, Hedges SB. 2009. A major clade of prokaryotes with ancient adaptations to life on land. *Mol Biol Evol* **26**:335–343.
doi:10.1093/molbev/msn247
- Benton MJ, Donoghue PCJ. 2007. Paleontological evidence to date the tree of life. *Mol Biol Evol*. doi:10.1093/molbev/msl150
- Berney C, Pawlowski J. 2006. A molecular time-scale for eukaryote evolution recalibrated with the continuous microfossil record. *Proc R Soc B Biol Sci* **273**:1867–1872. doi:10.1098/rspb.2006.3537
- Betts HC, Puttick MN, Clark JW, Williams TA, Donoghue PCJ, Pisani D. 2018. Integrated genomic and fossil evidence illuminates life's early evolution and eukaryote origin. *Nat Ecol Evol* **2**:1556–1562. doi:10.1038/s41559-018-0644-x
- Crowe SA, Døssing LN, Beukes NJ, Bau M, Kruger SJ, Frei R, Canfield DE. 2013. Atmospheric oxygenation three billion years ago. *Nature* **501**:535–538.
doi:10.1038/nature12426
- Dohrmann M, Wörheide G. 2017. Dating early animal evolution using phylogenomic data/631/181/735/631/181/414 article. *Sci Rep* **7**. doi:10.1038/s41598-017-03791-w
- Donoghue PCJ, Benton MJ. 2007. Rocks and clocks: calibrating the Tree of Life using fossils and molecules. *Trends Ecol Evol*. doi:10.1016/j.tree.2007.05.005
- Hug LA, Baker BJ, Anantharaman K, Brown CT, Probst AJ, Castelle CJ, Butterfield CN, Hermsdorf AW, Amano Y, Ise K, Suzuki Y, Dudek N, Relman DA, Finstad

- KM, Amundson R, Thomas BC, Banfield JF. 2016. A new view of the tree of life. *Nat Microbiol* **1**. doi:10.1038/nmicrobiol.2016.48
- Magnabosco C, Moore KR, Wolfe JM, Fournier GP. 2018. Dating phototrophic microbial lineages with reticulate gene histories. *Geobiology* **16**:179–189. doi:10.1111/gbi.12273
- Martijn J, Vosseberg J, Guy L, Offre P, Ettema TJG. 2018. Deep mitochondrial origin outside the sampled alphaproteobacteria. *Nature* **557**:101–105. doi:10.1038/s41586-018-0059-5
- Méheust R, Burstein D, Castelle CJ, Banfield JF. 2019. The distinction of CPR bacteria from other bacteria based on protein family content. *Nat Commun* **10**. doi:10.1038/s41467-019-12171-z
- Pittis AA, Gabaldón T. 2016. Late acquisition of mitochondria by a host with chimaeric prokaryotic ancestry. *Nature* **531**:101–104. doi:10.1038/nature16941
- Planavsky NJ, Asael D, Hofmann A, Reinhard CT, Lalonde S V, Knudsen A, Wang X, Ossa Ossa F, Pecoits E, Smith AJB, Beukes NJ, Bekker A, Johnson TM, Konhauser KO, Lyons TW, Rouxel OJ. 2014. Evidence for oxygenic photosynthesis half a billion years before the Great Oxidation Event. *Nat Geosci* **7**:283–286. doi:10.1038/ngeo2122
- Shih PM, Hemp J, Ward LM, Matzke NJ, Fischer WW. 2017. Crown group Oxyphotobacteria postdate the rise of oxygen. *Geobiology* **15**:19–29. doi:10.1111/gbi.12200
- Szöllősi GJ, Höhna S, Williams TA, Schrempf D, Daubin V, Boussau B. 2021. Relative time constraints improve molecular dating. *bioRxiv* 2020.10.17.343889. doi:10.1101/2020.10.17.343889
- Wang Z, Wu M. 2015. An integrated phylogenomic approach toward pinpointing the origin of mitochondria. *Sci Rep* **5**:7949. doi:10.1038/srep07949
- Zhang H, Sun Y, Zeng Q, Crowe SA, Luo H. 2021. Snowball Earths, population bottlenecks, and the evolution of marine photosynthetic bacteria. *bioRxiv*. doi:10.1101/2020.11.24.395392

Zimmer A, Lang D, Richardt S, Frank W, Reski R, Rensing SA. 2007. Dating the early evolution of plants: Detection and molecular clock analyses of orthologs. *Mol Genet Genomics* **278**:393–402. doi:10.1007/s00438-007-0257-6

Reviewers' Comments:

Reviewer #1:

Remarks to the Author:

In this revised version of their manuscript, the authors have carefully and sufficiently addressed my previous methodological comments. The revised analyses correctly employ fossil calibrations, sufficiently test root prior hypotheses, and have updated their figures, results, and discussion sections accordingly. The revised manuscript fairly identifies the sources of dating uncertainty in their analyses, and provides sufficient information for readers to interpret age estimates based on these specific parameters and assumptions. This has resulted in a much improved and informative paper.

Reviewer #2:

Remarks to the Author:

Wang and Luo have done an extensive revision of their work and have answered satisfactorily, within the limits of this kind of analysis, to my previous criticisms. I find the idea interesting, although I still see strong limitations in the available bacterial taxon sampling (no well identified alphaproteobacterial line ancestral to mitochondria). The authors could be more explicit in their discussion about how their dating might change should a closer lineage (within or out from known Alphaproteobacteria) to mitochondria be discovered.

How would their results/interpretation change in the case of mitochondria-early hypotheses for the origin of eukaryotes? This deserves a clear reference in the discussion.

Reviewer #3:

Remarks to the Author:

I appreciate the efforts made by the authors to address all the comments made. I do not necessarily agree with the analysis of the dependence of the estimates on the cyanobacteria root, especially when it is moved to 4500 MA (but a similar argument can be made for using 3000 MA as a minimum boundary). I think most would agree that this very old boundary for the root is not reasonable as cyanobacteria are not among the first groups to have evolved and 4500 is, effectively, the time of the origin of Earth. Thus, placing the root at 4500 means ignoring the evolutionary steps that occurred from the origin of life until cyanobacteria actually evolved. It is not surprising at all that if the boundaries are older, the time estimates will be older. This is a well known behavior of molecular clocks. I understand this analysis was made in response to another reviewer's comment so I do not fault the authors for carrying it out. It simply shows the lack of consensus on Cyanobacteria phylogenetic position. However, it still does not really justify the conclusion that the mitochondria-based method is better than the cyano-based. This conclusion seems to be mostly driven by the "side" the authors choose, whether the root maximum should be really old or not. Moreover, if extended, the argument made by the authors should be applied to most molecular clock analysis of prokaryotes because it is always possible to find more eukaryotic calibrations than prokaryotic ones. I do not disagree with the conclusions of the authors but I would like to see a more clear acknowledgment of the correlation between their conclusion (specifically that the mito-based method is better than the cyano-based one) with the scenario of cyanobacteria chosen. On a related note, if the mitochondria and cyanobacteria with GOE-based calibrations agree with each other, shouldn't this be validation for the validity of those calibrations instead of the deeper ones? This could settle the issue of what calibrations should be used if someone wishes to use the cyanobacteria-based method (or, otherwise, it could invalidate the mito-based method).

Beyond this point, the authors addressed other comments satisfactorily.

We thank the reviewers for their valuable comments. As follows we provide point-to-point responses. Please note that the line numbers correspond to the clean version of the resubmitted manuscript.

REVIEWERS' COMMENTS

Reviewer #1 (Remarks to the Author):

In this revised version of their manuscript, the authors have carefully and sufficiently addressed my previous methodological comments. The revised analyses correctly employ fossil calibrations, sufficiently test root prior hypotheses, and have updated their figures, results, and discussion sections accordingly. The revised manuscript fairly identifies the sources of dating uncertainty in their analyses, and provides sufficient information for readers to interpret age estimates based on these specific parameters and assumptions. This has resulted in a much improved and informative paper.

→We thank the reviewer for the positive comments.

Reviewer #2 (Remarks to the Author):

Wang and Luo have done an extensive revision of their work and have answered satisfactorily, within the limits of this kind of analysis, to my previous criticisms. I find the idea interesting, although I still see strong limitations in the available bacterial taxon sampling (no well identified alphaproteobacterial line ancestral to mitochondria). The authors could be more explicit in their discussion about how their dating might change should a closer lineage (within or out from known Alphaproteobacteria) to mitochondria be discovered.

How would their results/interpretation change in the case of mitochondria-early hypotheses for the origin of eukaryotes? This deserves a clear reference in the discussion.

→We have emphasized the potential uncertainty caused by the lack of a closer lineage to mitochondria in L288-L289.

Reviewer #3 (Remarks to the Author):

I appreciate the efforts made by the authors to address all the comments made. I do not necessarily agree with the analysis of the dependence of the estimates on the cyanobacteria root, especially when it is moved to 4500 MA (but a similar argument can be made for using 3000 MA as a minimum boundary). I think most would agree that this very old boundary for the root is not reasonable as cyanobacteria are not among the first groups to have evolved and 4500 is, effectively, the time of the origin of Earth. Thus, placing the root at 4500 means ignoring the evolutionary steps that occurred from the origin of life until cyanobacteria actually evolved. It is not surprising at all that if the boundaries are older, the time estimates will be older. This is a well known behavior of molecular clocks. I understand this analysis was made in response to another

reviewer's comment so I do not fault the authors for carrying it out. It simply shows the lack of consensus on Cyanobacteria phylogenetic position. However, it still does not really justify the conclusion that the mitochondria-based method is better than the cyano-based. This conclusion seems to be mostly driven by the "side" the authors choose, whether the root maximum should be really old or not. Moreover, if extended, the argument made by the authors should be applied to most molecular clock analysis of prokaryotes because it is always possible to find more eukaryotic calibrations than prokaryotic ones. I do not disagree with the conclusions of the authors but I would like to see a more clear acknowledgment of the correlation between their conclusion (specifically that the mito-based method is better than the cyano-based one) with the scenario of cyanobacteria chosen. On a related note, if the mitochondria and cyanobacteria with GOE-based calibrations agree with each other, shouldn't this be validation for the validity of those calibrations instead of the deeper ones? This could settle the issue of what calibrations should be used if someone wishes to use the cyanobacteria-based method (or, otherwise, it could invalidate the mito-based method).

Beyond this point, the authors addressed other comments satisfactorily.

→ We thank the reviewer for the constructive comments. In the revised manuscript, we have discussed the impact of the phylogenetic uncertainty of cyanobacteria on dating in L252-L257. We have additionally stated the importance of the cyanobacteria-based strategy for non-*Proteobacteria* lineages, and have given our suggestion for dating the bacterial tree of life in future studies in L241-L244 and L293-L299.